# The association between adult-life smoking and age-related cognitive decline in Danish men

Erik Lykke Mortensen[1,2]*, Kristine Hell[1,3], Gunhild Tidemann Okholm[1,4], Trine Flensborg-Madsen[1,4], Marie Grønkjær[1,5]

1 Unit of Medical Psychology, Section of Environmental Health, Department of Public Health, University of Copenhagen, Copenhagen, Denmark, 2 Center for Healthy Aging, University of Copenhagen, Copenhagen, Denmark, 3 The National Research Centre for the Working Environment, Copenhagen, Denmark, 4 National Institute of Public Health, University of Southern Denmark, Copenhagen, Denmark, 5 Center for Clinical Research and Prevention, Copenhagen University Hospital – Bispebjerg and Frederiksberg, Frederiksberg, Denmark

* elme@sund.ku.dk

## Abstract

### Background

Most previous studies of effects of smoking on age-related cognitive decline have compared cognitive decline in current smokers, former smokers, and never smokers rather than investigating the effects of pack-years. The aim of the present study was to analyze the association between smoking and age-related cognitive decline in a sample of men administered the same intelligence test in young adulthood and late midlife, using pack-years between the two assessments as the primary measure of exposure to smoking.

### Methods

In 5052 men, scores on a military intelligence test (BPP, Børge Priens Prøve) were available from young adulthood and a late midlife follow-up assessment including the same intelligence test and a comprehensive questionnaire on socio-demographic factors, lifestyle, and health. Information on smoking was self-reported at follow up for eight age periods, and pack-years were calculated from age 19 based on information on daily smoking and the duration of each age period. The differences in cognitive decline between adult-life smokers and non-smokers and the differences between light, moderate, and heavy smokers defined by pack-years were analyzed in linear regression models.

### Results

All smoking variables were only weakly associated with cognitive decline. Comparison of adult-life smokers and non-smokers showed less cognitive decline among smokers (1.12 IQ points, p < 0.001). Among smokers, analyses of pack-years suggested a weak dose-response relationship with more decline in heavy smokers than in light smokers (1.33 IQ points, p = 0.001). Independent of pack-years, current smoking was associated with larger cognitive decline than former smoking (1.73 IQ points, p < 0.001).

**Data availability statement:** The data that support the findings of this study are available from the authors but restrictions apply to the availability of these data. For the current study the data were used under license from the data inspection authorities at University of Copenhagen, and so are not publicly available. However, data are available from the authors upon reasonable request and with permission from relevant authorities. Queries regarding data access and more information about the cohort can be directed to Gunhild Tidemann Okholm [gunh@sdu.dk or guch@sund.ku.dk] or to the DanACo research team [bfh-fp-ckff-danaco@regionh.dk].

**Funding:** The data collection for the DanACo cohort was supported by the Innovation Fund Denmark, Health and Clinical Research (grant number 603-00520B) to ELM, the Independent Research Fund Denmark [grant number: 8020-00094B] to ELM, the Faculty of Health and Medical Sciences, University of Copenhagen to MG, Svend Andersen foundation, and Doctor Sofus Carl Emil Friis and wife Olga Doris Friis's foundation to GTO. Support for ongoing research based on the DanACo cohort by the Lundbeck foundation [grant number: R380-2021–1433] to GTO, Helsefonden [grant number: 22-B-0196] to GTO, and by the internal research funds of Frederiksberg and Bispebjerg hospitals to GTO. The funders had no role in the study design, data collection and analysis, decision to publish, or preparation of the manuscript.

**Competing interests:** The authors have declared that no competing interests exist.

## Conclusion

Smoking explained negligible fractions of the variance in cognitive decline, and thus our results did not indicate that smoking is a strong predictor of cognitive decline. The effects of pack-years suggest a relatively weak, possibly cumulative effect of smoking across the adult lifespan. The difference in decline between smokers and non-smokers may reflect participation bias and selective attrition at follow-up while the effects of current smoking may reflect either temporary effects of smoking or individual and life-style characteristics associated with continuation of smoking into late midlife.

## Introduction

Tobacco smoke is probably one of the most significant sources of toxic chemical exposure to humans [1] and smoking is a plausible risk factor for accelerated cognitive decline because of smoking-related respiratory and cardiovascular disease [2–3] and because of direct effects on the brain, possibly related to accelerated cortical thinning and decreased grey matter volume [4, 5]. The increased risk of cognitive decline associated with smoking was confirmed by an early meta-analysis [6], but a later meta-analysis was unable to confirm a significant association between smoking and cognitive decline [7]. However, most of the available studies were not designed with the primary objective to analyze effects of smoking on cognitive decline, and consequently there are large differences among studies with respect to age of participants, retest intervals, and cognitive assessment. Most recent, observational studies have shown significant associations between smoking and larger cognitive decline [8–11], although there are also negative [12] or inconsistent findings [13].

Several studies have analyzed samples over 60 years old and with relatively short retest intervals of about 2–5 years [8,11,14,15]. Other studies have analyzed participants who were considerably younger at the time of initial cognitive assessment and with longer retest intervals [9,16–18]. The studies have applied a wide range of cognitive tests from dementia screening instruments such as the Mini-Mental State Examination [14,15] to a range of cognitive tests assumed to assess specific cognitive abilities such as memory, speed of processing, and fluency (e.g., [11,13,18]). Some studies have derived global measures of cognition, but these measures are not necessarily comparable since they were derived from different sets of specific tests [10,11,13], and the studies based on the Moray House Test are among the few including a measure of general intelligence [9,16]. Thus, methodological differences among studies may contribute to explain between-study variance in results. High age at initial cognitive assessment may make it difficult to detect effects of smoking because other age- and health-related factors may influence cognition. Short retest intervals often show little decline and may not be sensitive to smoking during the retest interval. Different cognitive measures are likely to show different sensitivity to the effects of smoking. Global measures of cognition tend to be more reliable and may be more sensitive to effects of smoking, since smoking may be associated with global brain atrophy and with structural and biochemical abnormalities in several brain regions [19] as well as impairment in most neuropsychological functions [19,20].

Most studies have been able to divide the sample according to smoking status into current smokers, former smokers, and never smokers. The most consistent finding seems to be that current smokers show more decline than never smokers [6,10,11,15,16] and that former smokers have shown less decline than current smokers, and sometimes do not deviate significantly from never smokers [6,11,14,16]. When a distinction has been made between recent and long-term former smokers, decline in long-term former smokers has typically been

similar to decline in never smokers [10], suggesting that the cognitive effects of smoking are only temporary and wear off if smoking stops.

Smoking status at the time of the cognitive follow-up does not provide information that permits analysis of potential dose-response relationships between exposure to smoking and cognitive decline. However, in some studies, sufficient information has been available to calculate life-time pack-years and demonstrate a dose-response relationship [8,10,15], but other studies have found no association for pack-years when never-smokers were included in the analysis [9] or when the analysis only included smokers [18]. Typically, pack-years have been calculated to reflect life-time smoking, but since life-time smoking may influence initial cognitive performance in midlife or older age, it may be important to distinguish between smoking before initial cognitive assessment and smoking during the retest interval.

The primary aim of the present study was to analyze the association between adult-life smoking and age-related cognitive decline in a sample of men administered the same intelligence test in young adulthood and late midlife, using pack-years between the two assessments as the primary measure of exposure to smoking. Additional aims were to compare cognitive decline in smokers and non-smokers, to analyze effects of smoking before adulthood, and to analyze effects of current smoking at the midlife follow-up.

In studies of smoking and age-related cognitive decline, study design variables such as retest interval and age at follow-up should be included in statistical models to reduce associated variance in the observed cognitive decline. It is even more essential to incorporate potentially confounding variables such as young adult cognitive ability and educational attainment that may be associated with both smoking and cognitive decline. This is also the case for alcohol consumption and physical morbidity, but these variables may also mediate effects of smoking because smoking may influence alcohol consumption and is known to have adverse effects on health. For this reason, we only included alcohol consumption and physical morbidity as covariates in sensitivity analyses.

## Methods

### Study design

Detailed information on both smoking and cognitive ability was available from the DanACo cohort [21]. Additionally, the study is based on data from national health registries, obtained by using the unique personal identification number, which is assigned to all Danish residents [22]. Relevant information (i.e., address) from the national registries was updated frequently during data collection. When the participants were invited to participate, they received written information about the purpose of the study and the study procedures. Based on this information they decided whether to participate or not. In case of a positive decision, the participant booked a time for the follow-up assessment.

### Study population

The men included in the DanACo cohort participated in either the LiKO-15 study or the DiaKO-19 study [21]. The men invited to these studies were selected from two databases with draft-board information [23,24]. In total, 37,444 men were invited and 5340 men participated in a midlife follow-up study. Due to technical problems with the military computer system (n = 157) and missing data on smoking exposure or covariates (n = 131), data on 288 participants were excluded, resulting in a study sample comprising 5052 men born in 1949-61. The study included information from draft board examinations in young adulthood (1968-89), and from the follow-up examination in late midlife (LiKO-15 7th September 2015 – 7th June 2017 and DiaKO-19 16th September 2019 - 25th May 2022). The draft board examination

included assessment of school education, height, weight, and an intelligence test. The follow-up examination consisted of the same intelligence test and a comprehensive questionnaire on socio-demographic factors, lifestyle, and health.

## Measures

**Cognitive ability.** All Danish men, except those with disqualifying disease, are required to appear before a draft board when they turn 18 years old. Among other assessments, they complete an intelligence test called Børge Priens Prøve (BPP), which has been used by the Danish military for more than 50 years [25]. The BPP is a 45-minute intelligence test comprising 78 items in four subtests: letter matrices (19 items, 15 min), verbal analogies (24 items, 5 min), number series (17 items, 15 min), and geometric figures (18 items, 10 min). The BPP score (the number of correct answers summed across the four subtests) is highly correlated (r = .82) with the full-scale IQ of the Wechsler Adult Intelligence Scale [26]. The four subtests are highly inter-correlated and they all measure aspects of abstract reasoning and problem solving, reflecting fluid intelligence. Thus, the BPP should be sensitive to age-related changes in intelligence, and this has in fact been the case in several studies of cognitive ageing based on the DanACo cohort [21].

The participants completed a paper-and-pencil version of the test at the baseline draft board examination and a computerized version at follow-up. The BPP test scores were linearly transformed to an IQ scale with a sample mean of 100 and a standard deviation of 15 at baseline in young adulthood.

The difference between the draft board and the midlife follow-up IQ score was used as the outcome measure of age-related cognitive changes.

**Smoking.** Retrospective information on smoking was self-reported at follow up. The participants were asked about amount of daily smoking during the following eight age periods: < 15, 15-18, 19-25, 26-30, 31-40, 41-50, 51-60, > 60. Total pack-years (the number of cigarettes/units smoked per day divided by 20 and multiplied by the number of years of smoking) [10] were calculated based on information on daily smoking and the duration of each period. Smoking information was categorized into cigarettes, cigarillo, cigar and pipe smoking, and the calculation was based on the following tobacco equivalents: 1 cigarette = 1 unit, 1 Cigarillo = 3 units, 1 cigar = 5 units and 1 pipe = 3 units [27].

In the study sample, about 73% appeared before a draft board at age 20 or younger, and since we were particularly interested in effects of smoking during the interval between baseline and follow-up BPP administration, pack-years were calculated from age 19. Based on this measure of pack-years, the study sample was divided into subsamples of adult-life non-smokers (0 pack-years) and smokers (>0 pack-years). Pack-years had a skewed distribution and to analyze potential dose-response effects of smoking, we further divided the smokers into three categories: light smokers (0-20 pack-years), moderate smokers (>20-40 pack-years) and heavy smokers (>40 pack-years).

A binary variable indicated any reported smoking before age 19, and current smoking was indicated by a binary variable reflecting any smoking within the past 12 months.

## Covariates

Registry information on birthday, date of young adult and midlife intelligence testing was available and made it possible to calculate age at the follow-up assessment and the retest interval between the young and midlife assessments.

Data on admissions to Danish psychiatric wards from the Danish Psychiatric Central Research Register [28] were used to construct a binary indicator of psychiatric history, one or more admissions vs. no admissions (data available for 1969-2018).

At the follow-up examination, participants also reported information on education, and life-time alcohol consumption [21]. Based on information on both school and post-school vocational training and education, the total number of years of education was calculated.

Self-reported information on alcohol consumption was used to generate two adult-life alcohol consumption variables. The mean number of units of alcohol consumed per week across age periods from young adulthood (19 years or older) to late midlife and the number of years with adult-life weekly extreme binge drinking were included as continuous variables. Extreme binge drinking was defined as 10 units of alcohol or more on the same occasion [27].

Furthermore, information from the Danish National Patient Registry (available from 1977-2018) was used to calculate the Charlson Comorbidity Index (CCI). The CCI is a method of measuring burden of comorbidities [29].

## Statistical analyses

First, descriptive analyses were conducted on adult-life smokers and non-smokers. The differences between the two smoking categories were examined for smoking variables and covariates using $X^2$ test for categorical variables and t tests for continuous variables.

Second, the differences in IQ and IQ changes from baseline to follow-up were analyzed between adult-life smokers and non-smokers and among light, moderate, and heavy smokers. These analyses were conducted using mixed models including a group factor (smoking/non-smoking or light/moderate/heavy smokers), a time factor (baseline/follow-up) and a group x time interaction factor.

Third, the differences in IQ changes between adult-life smokers and non-smokers and the differences between light, moderate, and heavy smokers were further analyzed in linear regression models. Three regression models were analyzed. Model 1 investigated the association between smoking and IQ changes adjusting for retest interval, follow-up age, and psychiatric history, while model 2 additionally adjusted for young adult IQ scores, and model 3 included years of education in addition to the variables in model 2. Flynn effects according to year of birth are well-described for the BPP [30]. However, in the study sample, year of birth was not associated with young adult BPP scores, and consequently year of birth was not included in the statistical models.

The causal status of alcohol consumption and physical morbidity is ambiguous, since both factors may be influenced by smoking. Consequently, weekly alcohol consumption, years with weekly extreme binge drinking, and the Charlson Comorbidity Index were not included in the three main models, but were analyzed in separate sensitivity analyses.

The study sample included 1181 men who had been admitted to psychiatric wards, and since psychiatric disorders may be related to smoking and cognitive decline, the potential interactions on cognitive decline of smoking and psychiatric history were analyzed in preliminary analyses. No significant interactions were observed for the analyses of smokers vs. non-smokers or for the analyses of the three pack-year categories. However, this binary indicator variable was included in all statistical models, because the main effect of psychiatric history was significant in analyses based on the full sample and because of the literature on cognitive impairment in mental disorders [31].

Our measure of pack-years reflects consumption over the adult lifespan but does not distinguish between smoking in different age periods or between current and previous smoking. However, many studies have focused on current smoking vs. never or former smoking [6,7], and consequently we included a binary indicator of current smoking as well as an indicator of smoking before age 19 as our measure of pack-years only included smoking from age 19. Among smokers, these two indicator variables were analyzed together with the three-category

pack-year variable in a model with mutual adjustment and in models with the covariates in models 1-3.

Since current smoking or smoking before age 19 might modify the effects of pack-years on cognitive decline, preliminary analyses tested interactions of the three pack-year categories with current smoking and with smoking before age 19. None of the overall tests for interaction were significant, and the results will only be presented for the models including main effects of these variables.

Finally, the pack-year measure was analyzed as a continuous variable in a sensitivity analysis.

All analyses were performed using Stata 18.

## Ethics approval

All methods were carried out in accordance with relevant guidelines and regulations. Informed consent was obtained from all participants. The DanACo data-collection projects were submitted for ethics approval by the Committee on Health Research Ethics in the Capital region, but the Committee ruled that according to Danish law (Scientific Ethical Committees Act (in Danish: Komiteloven), article 14, paragraph 2) approval was not required as the studies did not involve collection of biological material.

## Results

As expected for this generation of Danish men, the majority of the sample (63.1%) had been smoking at some point during adult life (Table 1). The participants in the smoker category had an average of 27.41 pack-years, and 75% of the smokers were smoking before age 19 while only 26% were current smokers.

The non-smoking participants were slightly older at the draft board assessment while the adult-life smokers were slightly older at the follow-up, resulting in almost a one year longer

**Table 1. Characteristics of the study sample divided in adult-life non-smokers and smokers.**

|  | | Adult-life (from age 19) | | |
|---|---|---|---|---|
|  | Total | Smokers | Non-smokers | P value* |
|  | Mean (SD) | Mean (SD) | Mean (SD) | |
| N(%) | 5052 (100.00) | 3188 (63.10) | 1864 (36.90) | ------- |
| **Smoking variables** | | | | |
| Pack-years (19 and older) | 21.03 (24.54) | 27.41 (24.70) | 0.00 (0.00) | ------- |
| Smoker before age 19 (N(%)) | 2503 (49.54) | 2388 (74.91) | 115 (6.17) | <0.001 |
| Current smoker (N(%)) | 814 (16.11) | 814 (25.53) | 0 (0.00) | ------- |
| **Covariates** | | | | |
| Age at draft board assessment | 20.38 (2.11) | 20.29 (2.07) | 20.52 (2.18) | <0.001 |
| Age at follow-up | 64.37 (4.21) | 64.60 (4.14) | 63.99 (4.30) | <0.001 |
| Retest interval | 44.00 (4.33) | 44.30 (4.27) | 43.47 (4.38) | <0.001 |
| Years of education | 13.73 (2.51) | 13.48 (2.51) | 14.15 (2.46) | <0.001 |
| Years with weekly extreme binge drinking | 4.66 (10.81) | 5.73 (11.83) | 2.84 (8.52) | <0.001 |
| Adult-life weekly alcohol consumption | 12.31 (11.70) | 13.63 (13.18) | 10.05 (8.11) | <0.001 |
| Charlson Comorbidity Index score | 0.86 (1.55) | 0.95 (1.63) | 0.71 (1.41) | <0.001 |
| Psychiatric admission (N(%)) | 1181 (23.38) | 824 (25.85) | 357 (19.15) | <0.001 |

*P values refer to the significance of differences between smokers and non-smokers.

retest interval among smokers than among non-smokers. Compared to adult-life smokers, the non-smoking participants had significantly longer education (average 14.15 years vs. 13.48 years) and significantly lower Charlson Comorbidity Index (CCI) score (0.71 vs 0.95). Smoking was associated with significantly higher alcohol consumption and the number of years with weekly extreme binge drinking was also significantly and substantially higher in smokers than in non-smokers (5.73 years vs. 2.84 years).

Despite having the highest IQ score at both the draft board (102.64) and the follow-up examinations (95.47), the adult-life non-smokers had the largest decline in IQ scores (7.17) (Table 2). In contrast, the adult-life smokers had lower IQ scores (98.46 and 92.79), but also significantly smaller decline in IQ scores (5.67).

When dividing the 3188 smokers into light, moderate, and heavy smokers, about 45% were light smokers and about 23% were heavy smokers (Table 2). The light smokers had the highest baseline IQ score (101.07) compared with the moderate and heavy smokers (97.36 and 94.87, respectively) as well as the smallest decline in IQ scores (5.47) compared with the two other groups (5.54 for moderate smokers and 6.23 for heavy smokers). However, the overall test of these differences was not significant, and contrasts comparing the three groups pairwise also showed no significant differences in decline (the p value for the contrast between light and heavy smokers was 0.09).

Linear regression analyses of IQ decline confirmed the smaller decline among adult-life smokers than among non-smokers as positive coefficients indicate less decline (Table 3). The difference in mean decline was largest for model 1 and smaller for model 2 and model 3 with additional adjustment for young adult intelligence scores and years of education, but differences in decline were significant for all three models. The analyses of the three pack-year categories showed no significant differences for model 1, but significant differences for model 2 and model 3, although only the difference between light and heavy smokers was significant for model 3. However, compared with light smokers, the larger coefficients (indicating larger decline) observed for heavy smoking than for moderate smoking support a dose-response relationship among adult-life smokers.

**Table 2. Mean IQ scores at draft board baseline and midlife follow-up assessment for adult-life smokers and non-smokers, as well as light-, moderate-, and heavy-smokers\*.**

|  | N (%) | Baseline IQ Mean (SD) | Follow-up IQ Mean (SD) | Difference Mean (SD) | P value for change |
|---|---|---|---|---|---|
| Total Sample | 5052 (100.00) | 100.00 (15.00) | 93.78 (14.54) | -6.22 (9.67) | <0.001 |
| Smokers | 3188 (63.10) | 98.46 (14.87) | 92.79 (14.39) | -5.67 (9.81) | <0.001 |
| Non-smokers | 1864 (36.90) | 102.64 (14.85) | 95.47 (14.65) | -7.17 (9.36) | <0.001 |
| P value |  | <0.001[a] | <0.001[a] | <0.001[b] |  |
| Smokers only | N (%) | Baseline IQ Mean (SD) | Follow-up IQ Mean (SD) | Difference Mean (SD) | P value for change |
| Light smokers | 1434 (44.98) | 101.07 (13.97 | 95.60 (13.43) | -5.47 (9.45) | <0.001 |
| Moderate smokers | 1020 (31.99) | 97.36 (14.86) | 91.82 (14.29) | -5.54 (9.83) | <0.001 |
| Heavy smokers | 734 (23.02) | 94.87 (15.67) | 88.64 (15.16) | -6.23 (10.43) | <0.001 |
| P value |  | <0.001[a] | <0.001[a] | P = 0.205[b] |  |

*p values refer to mixed models including a group factor (smoking/non-smoking or light/moderate/heavy smokers), a time factor (baseline/follow-up) and a group x time interaction factor.

[a] p values refer to significance of differences between smokers and non-smokers and among light, moderate and heavy smokers.

[b] p values refer to the group x time interaction

Further, analyses of the adult-life smoking subsample showed that current smoking at the midlife follow-up was associated with more decline compared with non-smoking at follow-up while smoking before age 19 compared with non-smoking before age 19 was associated with less decline (Table 4). In models 2 and 3, including both these variables in addition to potential confounders and baseline IQ, the p-values remained statistically significant for current smoking and smoking before age 19 (p < 0.001).

The patterns of significance for the adult-life pack-year categories were similar for the models in Table 3 and Table 4 although the coefficients were slightly smaller when current smoking and smoking before age 19 were included.

In both Table 3 and 4, model 2 – adjusting for design variables (retest interval length, age at follow-up, and psychiatric history) and young adult intelligence – showed the largest coefficients for the pack-year categories, and the coefficients became somewhat smaller when further adjusted for years of educations.

In sensitivity analyses, the models in Tables 3 and 4 were repeated with additional adjustments for adult-life weekly alcohol consumption, years with weekly extreme binge drinking, and the Charlson Comorbidity Index-score (see Tables S1-S3 in appendix). These analyses

**Table 3. Associations of adult-life smoking status and three pack-year categories* with IQ change in linear regression analyses (N = 5052 and 3188 respectively)**.**

| | Model 1 | | | Model 2 | | | Model 3 | | |
|---|---|---|---|---|---|---|---|---|---|
| Predictor | Coef-ficient | CI | P value | Coef-ficient | CI | P value | Coef-ficient | CI | P value |
| Adult-life non-smoker | Ref. | | | Ref. | | | Ref. | | |
| Adult-life smoker | 1.73 | 1.19;2.27 | <0.001 | 0.95 | 0.44;1.46 | <0.001 | 1.12 | 0.61;1.62 | <0.001 |
| Adult-life smokers only | Coef-ficient | CI | P value | Coef-ficient | CI | P value | Coef-ficient | CI | P value |
| Light smoker | Ref. | | | Ref. | | | Ref. | | |
| Moderate smoker | -0.18 | -0.94;0.59 | 0.649 | -0.91 | -1.62;-0.19 | 0.013 | -0.60 | -1.31;0.12 | 0.100 |
| Heavy smoker | -0.60 | -1.45;0.26 | 0.173 | -1.81 | -2.62;-1.00 | <0.001 | -1.33 | -2.14;-0.52 | 0.001 |

*Based on pack-years of smoking from age 19 to midlife follow-up.

**Model 1: adjusted for retest interval length, age at follow-up, and psychiatric history. Model 2 additionally adjusted for young adult IQ scores, and model 3 additionally adjusted for years of educations.

**Table 4. Adjusted associations with IQ changes of three pack-year categories, current smoking, and smoking before age 19 among adult-life smokers (N = 3188)*.**

| | Model 1 | | | Model 2 | | | Model 3 | | |
|---|---|---|---|---|---|---|---|---|---|
| Adult-life smokers only | Coef-ficient | CI | P value | Coef-ficient | CI | P value | Coef-ficient | CI | P value |
| Light smoker | Ref. | | | Ref. | | | Ref. | | |
| Moderate smoker | -0.19 | -0.96;0.58 | 0.631 | -0.85 | -1.57;-0.12 | 0.022 | -0.55 | -1.27;0.17 | 0.133 |
| Heavy smoker | -0.36 | -1.26;0.55 | 0.438 | -1.47 | -2.32;-0.63 | 0.001 | -1.02 | -1.87;-0.17 | 0.019 |
| Not current Smoker | Ref. | | | Ref. | | | Ref. | | |
| Current smoker | -1.83 | -2.62;-1.03 | < 0.001 | -1.76 | -2.51;-1.02 | < 0.001 | -1.73 | -2.47;-0.99 | <0.001 |
| Not smoking before age 19 | Ref. | | | Ref. | | | Ref. | | |
| Smoking before age 19 | 2.08 | 1.32;2.85 | < 0.001 | 1.52 | 0.80;2.23 | < 0.001 | 1.69 | 0.98;2.41 | <0.001 |

*Reference groups are adult-life light smokers, current non-smoker, and not smoking before age 19. Model 1: Mutually adjusted for smoking variables and adjusted for retest interval length, age at follow-up, and psychiatric history. Model 2: additionally adjusted for young adult intelligence. Model 3: additionally adjusted for years of education.

**Table 5. Variance explained (R²) by covariates and smoking variables\*.**

| Statistical model: | Model 1 | Model 2 | Model 3 |
|---|---|---|---|
| Covariates | 6.76% | 18.61% | 20.14% |
| Three pack-year categories[1] | 0.05% | 0.51% | 0.26% |
| Three pack-year categories[2] | 0.02% | 0.32% | 0.14% |
| Current smoking | 0.58% | 0.54% | 0.52% |
| Smoking before age 19 | 0.82% | 0.43% | 0.54% |

\*Model 1: Mutually adjusted for smoking variables and adjusted for retest interval length, age at follow-up, and psychiatric history. Model 2: additionally adjusted for young adult intelligence. Model 3: additionally adjusted for years of education.

[1] Models with the three pack-year categories included as the only smoking factor.

[2] Models including the three pack-year categories as well as current smoking and smoking before age 19.

showed slightly larger positive coefficients for adult-life smoking vs. non-smoking and smaller negative coefficients when comparing light smokers to moderate and heavy smokers. These trends reflect the fact that heavy smokers have shorter education, consume more alcohol, and have more health problems than light smokers.

Table 5 shows the variance explained (R-squared) by covariates and smoking-related variables for the models in Tables 3 and 4. For model 2 and 3 including young adult intelligence, the covariates explained about 18-20% of the variance while pack-year categories and the other smoking variables all explained less than 1% of the variance. Thus, the smoking variables – even when they were statistically significant – were only weakly associated with age-related cognitive decline.

Among adult-life smokers, the analysis of the association between IQ changes and the continuous pack-year variable (Table S4 in appendix) corroborated the results for the three-category pack-year variable presented in Tables 3 and 4. Model 1 adjusting for retest interval, age at follow-up, and psychiatric history was non-significant, but models 2 and 3 including young adult intelligence and, in addition, years of education were significant. However, this association was quadratic, reflecting decreasing negative effects on cognitive decline with more pack-years at higher levels of smoking. The continuous pack-year variable explained 0.07%, 0.49% and 0.25% of the variance in the three models corresponding to the models shown in Table 3. Table S4 also shows that current smoking and smoking before age 19 were significant in all models that also included the continuous pack-year variable.

## Discussion

The present study analyzed effects of smoking on cognitive decline over four decades from young adulthood to late midlife in a relatively large sample of men born 1949-61. Using total pack-years from age 19 to the midlife follow-up as measure of exposure, less cognitive decline was observed in smokers than in non-smokers, but among smokers more decline was observed in heavy smokers than in light smokers. Current smoking at the time of follow-up was associated with more decline while smoking before age 19 was associated with less cognitive decline. However, irrespective of statistical significance, all smoking variables were only weakly associated with age-related cognitive decline.

### Smokers vs. non-smokers

The finding of less decline in smokers compared with non-smokers raises several questions. Most studies finding effects of smoking on cognitive decline have compared current smokers with never smokers [6,10,11,16]. In contrast to smoking status at the time of follow-up,

our definition of smokers and non-smokers was based on pack-years from young adulthood (around age 19) to late midlife. Thus, we included men who only smoked during a period in young adulthood among smokers and the few men (n = 115) who only smoked before age 19 among non-smokers. In other studies, these men would be included among former smokers, and typically, associations with cognitive decline have been weaker or non-significant among former smokers [6,10,11,16].

It remains to be explained why adult-life smokers apparently show less cognitive decline in the present study. There is evidence that nicotine may have positive acute effects on cognition, particularly some aspects of attention and memory but these are short-term effects, which may play a role in withdrawal symptoms [32,33] and it is important to distinguish acute effects of nicotine from the effects of chronic exposure to the many compounds in tobacco smoke [19]. A more likely explanation may be related to participation bias because long-term smoking influences morbidity and mortality and because smokers may be less motivated to participate in health-related cognitive studies [7,34]. To the extent that non-participation due to these factors is related to cognitive decline, selective attrition among smokers may contribute to the difference in decline between smokers and non-smokers – particularly since the overall participation rate was only 14.3% [21]. However, Table 1 shows that non-smokers had longer education, drank less alcohol, had lower Charlson Comorbidity Index-score and fewer psychiatric admissions. These are all characteristics that are expected to be associated with less cognitive decline, but a factor contributing to more decline in non-smokers may be the significantly higher young adult IQ in non-smokers (the mean difference corresponded to about 0.3 of the theoretical standard deviation of 15), as higher baseline IQ has been shown to be associated with more cognitive decline [35] in the LiKO-15 subsample of DanACo and in other Danish cohorts [36]. The correlation between baseline IQ and cognitive decline was 0.37 in our study, but the difference between smokers and non-smokers remained significant when adjustment for baseline IQ was included in the statistical models. It is, however, possible that both participation bias and young adult intelligence scores contribute to the difference in decline between smokers and non-smokers.

Other studies based on draft board data have also observed lower young adult IQ in smokers than among never smokers [37,38]. This could potentially reflect early effects of smoking [39], but several studies have observed lower childhood IQ among future smokers [16,40,41], suggesting that low cognitive ability may be a risk factor for initiating smoking and that the many cross-sectional studies of cognitive effects of smoking [19] should be interpreted with caution.

## Effects of pack-years

Significant differences among the three adult-life pack-year categories were only observed in models adjusting for young adult intelligence. These models showed significantly more decline for heavy smokers compared with light smokers, while moderate smokers only differed significantly from light smokers when not adjusting for education or current smoking. However, for all models, the negative coefficients were smaller for moderate smokers than heavy smokers, and in this sense, the results suggest a dose-response relationship between smoking defined by pack-years and cognitive decline among smokers. A dose-response relationship for pack-years has previously been observed [8,10], but the rather small regression coefficients and explained variance in our study suggest relatively weak associations, and other studies have found no association for pack-years [9]. The latter study used pack-years as a continuous measure, and the association with cognitive decline became non-significant in analyses adjusting for adult social class and childhood intelligence. In contrast, the association between the three-category pack-year variable and cognitive decline remained significant

in analyses adjusted for both young adult intelligence and education, although measures of education are often more strongly associated with cognitive outcomes than broader measures of social class [42]. A difference between the two studies is that Corley et al. [9] included non-smokers in the analysis of pack-years, while we – like Sabia et al. [18] – only analyzed smokers.

## Current smoking

Current smoking explains more variance in cognitive decline than the variance explained by pack-years, but the explained variance also suggests weak associations for all included smoking variables. This is corroborated by other studies, although these studies analyzed current smoking and pack-years in separate statistical models [8,9]. While many studies demonstrate effects of current smoking on cognitive decline, our results raise the issue of why current smoking is a significant predictor when a measure of amount of smoking such as pack-years is included in the models. A possible explanation is that pack-year measures primarily reflect the cumulative effects of life-time smoking, and that there are independent temporary or acute negative effects of current smoking. Although most studies report positive acute effects of nicotine or smoking [32,33], these effects may be age and health dependent, and an intervention study actually demonstrated positive effects of not smoking in 18 months in a sample comprising 68 years old or older people [43]. Another important factor is that the pack-year measure does not take age at smoking periods into account, and this is important if the relevant effects of smoking on the brain and body health are only temporary and wear off when smoking stops (as would be the case for men smoking heavily in young adulthood, but not in older age). In fact, the studies showing less decline in former smokers than current smokers [6,11,14,16] corroborate the assumption of temporary effects of smoking on cognition and may also contribute to explaining the weak effects of smoking in our study sample, comprising a majority of former smokers. In post-hoc analyses, we found that current smokers and men smoking in their 60s showed significantly larger cognitive decline than men who stopped smoking at an earlier age.

Table 1 shows that of all the 3188 adult-life smokers, only about 26% were current smokers at the time of the midlife follow-up. This indicates that quitting is normal in the sense that in this study sample, 7 out of 10 smoking men eventually quit, and suggests that current smokers comprise a selected group. In fact, t tests showed that current smokers had less favorable characteristics than former smokers with respect to education, young adult IQ, alcohol consumption, and comorbidity. Current smokers also had a substantially higher number of pack-years, but the effect of current smoking explained more variance than pack-years in both the unadjusted and adjusted statistical models. However, many other individual characteristics and life-style factors could contribute to the association between current smoking and cognitive decline, and differences between current and former smokers may become larger as more and more smokers give up smoking.

## Smoking before adulthood

Smoking before age 19 also explains more variance in cognitive decline than pack-years. The association between smoking before age 19 and less cognitive decline is unlikely to be a direct effect of smoking, but smoking before age 19 was also associated with less education and lower young adult intelligence as well as more morbidity. Lower intelligence in young adulthood was associated with less decline, but the effect of smoking before age 19 remained significant when adjusted for this variable and other covariates, suggesting that other individual characteristics and life-style factors not included in the statistical models contribute to the association.

## Strengths and limitations

In addition to the relatively large sample, the present study has several strengths: The assessment of intelligence in young adulthood and the use of the same test of intelligence at the midlife follow-up taking place after several decades is unique compared to other studies. The BPP intelligence test is essentially a test of fluid intelligence [44,45] and a one-factor model can be fitted to the four subtests [21]. While any effects of smoking on global cognition are important, the use of a uni-dimensional global measure may make it difficult to compare our results with studies analyzing more specific cognitive effects of smoking using tests of learning, memory, attention, and psychomotor speed [9,11].

The assessment of smoking was retrospective at midlife and thus susceptible to both recall errors and recall bias. However, smoking habits may be relatively stable for many men [46], and the detailed smoking information made it possible to calculate pack-years and to some extent to distinguish between smoking before and after the young adult intelligence assessment. However, since the assessment of smoking was only by 5- and 10-year intervals, our measure of pack-years since age 19 should only be considered an approximate measure of smoking after the young adult assessment of intelligence. In fact, it will overestimate the number of pack-years for men appearing before the draft board at age 20 or later, but statistical analyses showed no interaction between age at appearing before a draft board and effects of pack-years. Studies comparing prospective and retrospective assessment of pack-years are rare, but one study categorizing prospective and retrospective measures of pack-years found Cohen's kappa to be 0.79, demonstrating substantial agreement between the two types of data collection [47],

A particularly important weakness of the present study is the lack of information on the smoking habits of the majority of invited men who chose not to participate in the study. As discussed above, participation bias may have influenced the comparison of cognitive decline in smokers and non-smokers, but in fact also the comparisons of light, moderate, and heavy smokers. Thus, there is evidence that the participants in the DanACo cohort comprise a selected group with relatively high education and high intelligence in young adulthood [21]. Since smoking is related to both lower education and lower young adult intelligence, smoking may also be associated with lower participation, and this may particularly be the case for current smokers/heavy smokers with possible cognitive decline.

Our sample was selected to include a relatively large proportion of men with previous psychiatric hospital admissions (about 23% of the study sample), and this might have influenced the estimated effects of smoking. However, separate sensitivity analyses showed no significant interaction of psychiatric admission with either the smoker vs. non-smoker variable or the three-category pack-year variable, suggesting that the estimates of smoking on cognitive decline were not significantly influenced by this selection factor.

An important limitation is that the study sample consisted exclusively of men because the study was based on a draft board intelligence test. Since some studies have found sex differences in the effects of smoking on cognitive decline [10,48], it is an open question whether our results can be generalized to women.

Finally, the issue of residual and unmeasured confounding is particularly important for studies with unexpected results such as our comparison of smokers vs. non-smokers. Although the statistical models included several potentially confounding variables, differences in cognitive decline between smoking categories may reflect unmeasured individual characteristics and life-style factors.

## Conclusion

In this study based on administration of the same intelligence test in young adulthood and late midlife, smoking variables explained negligible fractions of the variance in cognitive decline, and thus our results did not indicate that smoking is a strong predictor of cognitive decline. Although non-smokers had the highest IQ score at both baseline and follow-up, comparison of adult-life smokers and non-smokers showed significantly less cognitive decline among smokers, which may reflect participation bias and selective attrition at follow-up.

Among adult-life smokers, analyses of adult-life pack-years suggested a weak, but possible dose-response relationship with more cognitive decline in heavy smokers than in light smokers. Current smoking explained more variance in cognitive decline than pack-years in models including both factors, suggesting temporary effects of smoking or individual and life-style characteristics associated with continuation of smoking into late midlife.

The observed weak associations between smoking and age-related cognitive decline suggest that other health effects of smoking are more important from a public health perspective.

## Supporting information

**S1 Appendix. Tables S1-S4.**
(DOCX)

## Acknowledgments

The authors would like to thank the Danish Defense for the permission to use the military intelligence test in the follow-up examinations and the personnel at the Military Recruitment and Career – Selection and Assessment Unit for excellent collaboration during the data collections. Moreover, the authors would like to thank the project workers including project coordinators and data collectors for their invaluable work in conducting the data collections for the DanACo cohorts. Finally, the authors thank all men who offered their time and participated in the follow-up examinations.

## Author contributions

**Conceptualization:** Erik Lykke Mortensen, Kristine Hell, Gunhild Tidemann Okholm, Trine Flensborg-Madsen, Marie Grønkjær.

**Data curation:** Erik Lykke Mortensen, Gunhild Tidemann Okholm, Marie Grønkjær.

**Formal analysis:** Erik Lykke Mortensen, Kristine Hell.

**Funding acquisition:** Erik Lykke Mortensen, Gunhild Tidemann Okholm, Marie Grønkjær.

**Investigation:** Erik Lykke Mortensen, Kristine Hell, Gunhild Tidemann Okholm, Trine Flensborg-Madsen, Marie Grønkjær.

**Methodology:** Erik Lykke Mortensen, Kristine Hell, Gunhild Tidemann Okholm, Trine Flensborg-Madsen, Marie Grønkjær.

**Project administration:** Erik Lykke Mortensen, Gunhild Tidemann Okholm, Marie Grønkjær.

**Resources:** Erik Lykke Mortensen, Gunhild Tidemann Okholm, Marie Grønkjær.

**Supervision:** Erik Lykke Mortensen, Gunhild Tidemann Okholm, Trine Flensborg-Madsen, Marie Grønkjær.

**Writing – original draft:** Erik Lykke Mortensen, Kristine Hell.

**Writing – review & editing:** Erik Lykke Mortensen, Kristine Hell, Gunhild Tidemann Okholm, Trine Flensborg-Madsen, Marie Grønkjær.

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
