## [Decision Letter · Decision Letter 0]

27 Nov 2024

PONE-D-24-37264The association between adult-life smoking and age-related cognitive decline in Danish menPLOS ONE

Dear Dr. Mortensen,

Thank you for submitting your manuscript to PLOS ONE. After careful consideration, we feel that it has merit but does not fully meet PLOS ONE’s publication criteria as it currently stands. Therefore, we invite you to submit a revised version of the manuscript that addresses the points raised during the review process.

The manuscript is well-written and methodologically strong and contribute additional knowledge on association between adult life smoking and age-related cognitive decline. The topic is important yet less studied and the details presented in this study is commendable. There are few issues that require clarification, and potential correction, to make this paper publishable.

We look forward to receiving your revised manuscript.

Kind regards,

Prakash K.C., Ph.D.

Academic Editor

PLOS ONE

Additional Editor Comments:

*Introduction: description of importance of other potential factors and covariates is missing, please include few sentences to justify the selection of covariates, why they were chosen?

*Results: Sensitivity analysis: there is an explanation in results section, but data/ result is missing, just explanation is not enough?

*Results: “Among adult life smokers the analysis of association between IQ changes and continuous pack years variable corroborated the results for the three-category pack-year variables in Table 3”, please present the data to support this, - statement such as data not shown is arbitrary

*Please revise and present just the most important results in conclusion, now the conclusion reads like discussion and may be misleading.

*A general remark: in conclusion: you do not need to endorse that the manuscript is methodologically strong leave it for readers to decide.

Reviewers' comments:

Reviewer's Responses to Questions

**Comments to the Author**

1. Is the manuscript technically sound, and do the data support the conclusions?

Reviewer #1: Yes

2. Has the statistical analysis been performed appropriately and rigorously? 

Reviewer #1: Yes

3. Have the authors made all data underlying the findings in their manuscript fully available?

Reviewer #1: Yes

4. Is the manuscript presented in an intelligible fashion and written in standard English?

Reviewer #1: Yes

5. Review Comments to the Author

Reviewer #1: This is a very relevant and interesting manuscript and provided information that may be useful. Please incorporate or address following comments.

Abstract:

Page 3, Line 4 – can you remove 'methodologically strong', no need to mention this.

Page 3, line 4-5 – ' and thus our results did not corroborate the hypothesis that smoking is a strong predictor of cognitive decline 'seems also unnecessary as no such hypothesis is mentioned in the whole article except in your conclusion.

Introduction:

Page 6, line 1-4, can you also mention other secondary objectives for which you did analysis and included in conclusion?

Can you add a sentence regarding biological or vascular mechanism/explanation on how smoking leads to cognitive decline?

Methods:

Page 6, Line 19: Can you specify the technical problem clearly, so that the reader can evaluate whether it affects the study results or not?

Page 7, Line 6-16, Can you add a-sentence information regarding validity of the tool for adult men and regarding its validation?

Factors such as alcohol consumption and education could influence cognitive decline and may be more important predictors than smoking. So it seems difficult to assess whether residual confounding could explain the weak associations. So was there any provision in your study to address residual confounding, please mention

Results:

Page 12, Line 10 – 'Table 1. Characteristics of the study sample divided in adult-life non-smokers and smokers': Some p values seem very not necessary to mention, with counts 0 in non-smokers, please edit accordingly.

Your table 2 is very hard to follow, please if possible, simplify the table formatting or break the information in two tables for better readability.

Discussion:

Limitation of self reported smoking and alcohol information in follow up – chances of recall and participant bias is discussed but can you discuss its relation to study results.

The statistical significance is not the indication of public health significance, you need to discuss whether this difference has any public health significance or not.

If smoking has a cumulative effect, we expect former smokers to experience more decline than the never-smokers, but the larger effect in current smokers suggests may be acute effects of smoking rather than a cumulative lifelong exposure. This is important to be included in your discussion.

6. PLOS authors have the option to publish the peer review history of their article (what does this mean? ). If published, this will include your full peer review and any attached files.

**Do you want your identity to be public for this peer review?** For information about this choice, including consent withdrawal, please see our Privacy Policy .

Reviewer #1: No

---

## [Author Response · Author response to Decision Letter 0]

2 Jan 2025

PONE-D-24-37264

The association between adult-life smoking and age-related cognitive decline in Danish men

PLOS ONE

Editor comments

Thank you for submitting your manuscript to PLOS ONE. After careful consideration, we feel that it has merit but does not fully meet PLOS ONE’s publication criteria as it currently stands. Therefore, we invite you to submit a revised version of the manuscript that addresses the points raised during the review process.

The manuscript is well-written and methodologically strong and contribute additional knowledge on association between adult life smoking and age-related cognitive decline. The topic is important yet less studied and the details presented in this study is commendable. There are few issues that require clarification, and potential correction, to make this paper publishable.

Author response: Thank you, we appreciate your positive evaluation of our manuscript.

Additional Editor Comments:

Editor Comment: Introduction: description of importance of other potential factors and covariates is missing, please include few sentences to justify the selection of covariates, why they were chosen?

Author response: We have inserted the following paragraph at the end of the introduction:

“In studies of smoking and age-related cognitive decline, study design variables such as retest interval and age at follow-up should be included in statistical models to reduce associated variance in the observed cognitive decline. It is even more essential to incorporate potentially confounding variables such as young adult cognitive ability and educational attainment that may be associated with both smoking and cognitive decline. This is also the case for alcohol consumption and physical morbidity, but these variables may also mediate effects of smoking because smoking may influence alcohol consumption and is known to have adverse effects on health. For this reason, we only included alcohol consumption and physical morbidity as covariates in sensitivity analyses.”

Editor Comment: Results: Sensitivity analysis: there is an explanation in results section, but data/ result is missing, just explanation is not enough?

Author response: We have decided to provide tables with the sensitivity analyses as supporting information. These tables are all included in the “appendix” file. Should the editor prefer any of the tables in the manuscript, they can of course be moved.

Editor Comment: Results: “Among adult life smokers the analysis of association between IQ changes and continuous pack years variable corroborated the results for the three-category pack-year variables in Table 3”, please present the data to support this, - statement such as data not shown is arbitrary

Author response: Inspired by this comment, we have included in the appendix a new table S4 showing the main results of pack-years as a continuous variable. Should the editor find that the table should be in the manuscript, it can of course be included as a new Table 6.

Editor Comment: Please revise and present just the most important results in conclusion, now the conclusion reads like discussion and may be misleading.

Author response: We have revised and shortened the conclusion, but have incorporated the suggestion by the reviewer that a statement on public health implications should be included. We found it natural to insert such a statement at the end of the concluding paragraph: “In this study based on administration of the same intelligence test in young adulthood and late midlife, smoking variables explained negligible fractions of the variance in cognitive decline, and thus our results did not indicate that smoking is a strong predictor of cognitive decline. Although non-smokers had the highest IQ score at both baseline and follow-up, comparison of adult-life smokers and non-smokers showed significantly less cognitive decline among smokers, which may reflect participation bias and selective attrition at follow-up.

Among adult-life smokers, analyses of adult-life pack-years suggested a weak, but possible dose-response relationship with more cognitive decline in heavy smokers than in light smokers. Current smoking explained more variance in cognitive decline than pack-years in models including both factors, suggesting temporary effects of smoking or individual and life-style characteristics associated with continuation of smoking into late midlife.

The observed weak associations between smoking and age-related cognitive decline suggest that other health effects of smoking are more important from a public health perspective.”

Editor Comment: A general remark: in conclusion: you do not need to endorse that the manuscript is methodologically strong leave it for readers to decide.

Author response: “methodologically strong” has been deleted.

Reviewer comments to the Author

1. Is the manuscript technically sound, and do the data support the conclusions?

Reviewer #1: Yes

2. Has the statistical analysis been performed appropriately and rigorously?

Reviewer #1: Yes

3. Have the authors made all data underlying the findings in their manuscript fully available?

Reviewer #1: Yes

4. Is the manuscript presented in an intelligible fashion and written in standard English?

Reviewer #1: Yes

5. Review Comments to the Author

Reviewer #1: This is a very relevant and interesting manuscript and provided information that may be useful. Please incorporate or address following comments.

Author response: Thank you for the positive evaluation of our manuscript.

Abstract:

Review: Page 3, Line 4 – can you remove 'methodologically strong', no need to mention this.

Author response: We have deleted 'methodologically strong'.

Review: Page 3, line 4-5 – ' and thus our results did not corroborate the hypothesis that smoking is a strong predictor of cognitive decline 'seems also unnecessary as no such hypothesis is mention,ed in the whole article except in your conclusion.

Author response: We have adjusted the text to “Smoking variables explained negligible fractions of the variance in cognitive decline, and thus our results did not indicate that smoking is a strong predictor of cognitive decline”. We have made the same change in the conclusion paragraph of the manuscript.

Introduction:

Review: Page 6, line 1-4, can you also mention other secondary objectives for which you did analysis and included in conclusion?

Author response: We have adjusted the paragraph to include a description of the additional aims:

“The primary aim of the present study was to analyze the association between adult-life smoking and age-related cognitive decline in a sample of men administered the same intelligence test in young adulthood and late midlife, using pack-years between the two assessments as the primary measure of exposure to smoking. Additional aims were to compare cognitive decline in smokers and non-smokers, to analyze effects of smoking before adulthood, and to analyze effects of current smoking at the midlife follow-up.”

Review: Can you add a sentence regarding biological or vascular mechanism/explanation on how smoking leads to cognitive decline?

Author response: The reviewer suggests to add a sentence on biological or vascular mechanisms, and we have elaborated the first paragraph of the introduction accordingly (the added sentence is underlined):

“Tobacco smoke is probably one of the most significant sources of toxic chemical exposure to humans [1] and smoking is a plausible risk factor for accelerated cognitive decline because of smoking-related respiratory and cardiovascular disease [2-3] and because of direct effects on the brain, possibly related to accelerated cortical thinning and decreased grey matter volume [4-5].”

Methods:

Review: Page 6, Line 19: Can you specify the technical problem clearly, so that the reader can evaluate whether it affects the study results or not?

Author response: At the late midlife follow-up the BPP intelligence test was administered by computer and the scores were collected at a server belonging to the Danish military. The loss of data was caused by technical problems with the military computer system, and thus completely unrelated to the cognitive performance of the participants. We have expanded the text in the manuscript accordingly:

“Due to technical problems with the military computer system (n = 157)”.

Review: Page 7, Line 6-16, Can you add a-sentence information regarding validity of the tool for adult men and regarding its validation?

In the manuscript we refer to Tom Teasdale’s review of the use of the BPP (Reference 25). The important point is that the test was specifically constructed for use by the Danish draft board to evaluate the intelligence of young men. The BPP has been used for this purpose for more than 50 years, and the military has not found it necessary to revise or adjust the test content. This is because the items of the test do not require specific knowledge, and they primarily assess fluid intelligence. No ceiling or floor effects have been observed, and this is also the case when the BPP has been administered to late midlife adult men The four subtests are highly inter-correlated and they all measure aspects of abstract reasoning and problem solving, reflecting fluid intelligence. Thus, the BPP should be sensitive to age-related changes in intelligence, and this has in fact been the case in several studies of cognitive ageing based on the DanACo cohort (see reference 21).

In the manuscript we also refer to a study demonstrating a correlation of 0.82 between BPP scores and the Wechsler Adult Intelligence Scale (WAIS), which is the most widely used individual test of intelligence (Reference 26). Thus, this correlation should be a sufficient indicator of the quality of the BPP, but to accommodate the reviewer’s concern we have inserted a sentence describing the use of the BPP by the Danish military:

“Among other assessments, they complete an intelligence test called Børge Priens Prøve (BPP), which has been used by the Danish military for more than 50 years [25]”.

We have also added the following sentences to the description of the BPP: “The four subtests are highly inter-correlated and they all measure aspects of abstract reasoning and problem solving, reflecting fluid intelligence. Thus, the BPP should be sensitive to age-related changes in intelligence, and this has in fact been the case in several studies of cognitive ageing based on the DanACo cohort [21].”

Review: Factors such as alcohol consumption and education could influence cognitive decline and may be more important predictors than smoking. So it seems difficult to assess whether residual confounding could explain the weak associations. So was there any provision in your study to address residual confounding, please mention

Author response: In tables 3, 4 and 5, years of education was included in model 3 of the statistical analyses. A previous study has shown relatively small effects of adult alcohol consumption (see reference 27), but based on these results both weekly alcohol consumption and years with extreme binge drinking were included in sensitivity analyses. This is described in the statistical analysis section and the results section. The reason that alcohol consumption was not included in the main analyses is that alcohol consumption (and the Charlson Comorbidity Index) may be influenced by smoking, and thus may be mediators of the association rather than confounders.

However, we agree with the reviewer that residual confounding may be an important factor in explaining some of the results. This is the reason why we refer to “individual characteristics and life-style factors” in the discussion, and to accommodate the reviewer’s concern we have inserted the following paragraph at the end of the strengths and limitations section:

“Finally, the issue of residual and unmeasured confounding is particularly important for studies with unexpected results such as our comparison of smokers and non-smokers. Although the statistical models included several potentially confounding variables, differences in cognitive decline between smoking categories may reflect unmeasured individual characteristics and life-style factors.”

Results:

Review: Page 12, Line 10 – 'Table 1. Characteristics of the study sample divided in adult-life non-smokers and smokers': Some p values seem very not necessary to mention, with counts 0 in non-smokers, please edit accordingly.

Author response: We have deleted the p values for pack-years and current smoker in table 1.

Review: Your table 2 is very hard to follow, please if possible, simplify the table formatting or break the information in two tables for better readability.

Author response: We have adjusted the format of table 2. It should be more readable now.

Discussion:

Limitation of self reported smoking and alcohol information in follow up – chances of recall and participant bias is discussed but can you discuss its relation to study results.

Author response: The discussion of smokers vs. non-smokers includes a discussion of participation bias in relation to the finding of less decline in smokers: “A more likely explanation may be related to participation bias because long-term smoking influences morbidity and mortality and because smokers may be less motivated to participate in health-related cognitive studies [7, 34]. To the extent that non-participation due to these factors is related to cognitive decline, selective attrition among smokers may contribute to the difference in decline between smokers and non-smokers – particularly since the overall participation rate was only 14.3% [21]. However, Table 1 shows that non-smokers had longer education, drank less alcohol, had lower Charlson Comorbidity Index-score and fewer psychiatric admissions. These are all characteristics that are expected to be associated with less cognitive decline, but a factor contributing to more decline in non-smokers may be the significantly higher young adult IQ in non-smokers (the mean difference corresponded to about 0.3 of the theoretical standard deviation of 15), as higher baseline IQ has been shown to be associated with more cognitive decline [35] in the LiKO-15 subsample of DanACo and in other Danish cohorts [36].”

In principle, similar concerns could be raised for the analyses of pack-years, current smoking and smoking before adulthood. However, in our opinion too many arguments would be repeated and consequently we have refrained from including repeated discussions of the influence of participation bias. Should the editor disagree with this decision, more discussion of participation bias

---

## [Editor Report · Decision Letter 1]

9 Feb 2025

The association between adult-life smoking and age-related cognitive decline in Danish men

PONE-D-24-37264R1

Dear Dr. Mortensen,

We’re pleased to inform you that your manuscript has been judged scientifically suitable for publication and will be formally accepted for publication once it meets all outstanding technical requirements.

Kind regards,

Prakash K.C., Ph.D.

Academic Editor

PLOS ONE
---

## [Editor Report · Acceptance letter]

PONE-D-24-37264R1

PLOS ONE

Dear Dr. Mortensen,

I'm pleased to inform you that your manuscript has been deemed suitable for publication in PLOS ONE. Congratulations! Your manuscript is now being handed over to our production team.

Kind regards,

on behalf of

Dr. Prakash K.C.

Academic Editor

PLOS ONE